# Sparse Flows: Pruning Continuous-depth Models

**Lucas Liebenwein**[*]
MIT CSAIL
lucas@csail.mit.edu

**Ramin Hasani**[*]
MIT CSAIL
rhasani@mit.edu

**Alexander Amini**
MIT CSAIL
amini@mit.edu

**Daniela Rus**
MIT CSAIL
rus@csail.mit.edu

## Abstract

Continuous deep learning architectures enable learning of flexible probabilistic models for predictive modeling as neural ordinary differential equations (ODEs), and for generative modeling as continuous normalizing flows. In this work, we design a framework to decipher the internal dynamics of these continuous depth models by pruning their network architectures. Our empirical results suggest that pruning improves generalization for neural ODEs in generative modeling. We empirically show that the improvement is because pruning helps avoid mode-collapse and flatten the loss surface. Moreover, pruning finds efficient neural ODE representations with up to 98% less parameters compared to the original network, without loss of accuracy. We hope our results will invigorate further research into the performance-size trade-offs of modern continuous-depth models.

## 1 Introduction

The continuous analog of normalizing flows (CNFs) (Chen et al., 2018) efficiently (Grathwohl et al., 2019) maps a latent space to data by ordinary differential equations (ODEs), relaxing the strong constraints over *discrete* normalizing flows (Dinh et al., 2016, Durkan et al., 2019, Huang et al., 2020, Kingma and Dhariwal, 2018, Papamakarios et al., 2017, Rezende and Mohamed, 2015). CNFs enable learning flexible models by unconstrained neural networks. While recent works investigated ways to improve CNFs' efficiency (Finlay et al., 2020, Grathwohl et al., 2019, Li et al., 2020), regularize the flows (Onken et al., 2020, Yang and

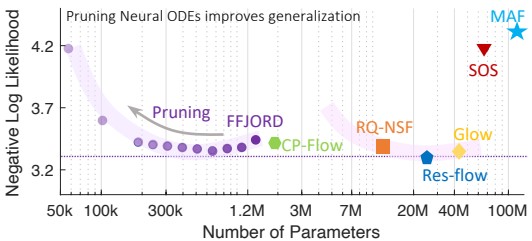

Figure 1: Pruning Neural ODEs improves their generalization with at least 1 order of magnitude less parameters. CIFAR-10 density estimation. Values and methods are described in Table 3.

Karniadakis, 2020), or solving their shortcomings such as crossing trajectories (Dupont et al., 2019, Massaroli et al., 2020), less is understood about their inner dynamics during and post training.

In this paper, we set out to use standard pruning algorithms to investigate generalization properties of sparse neural ODEs and continuous normalizing flows. In particular, we investigate how the inner dynamics and the modeling performance of a continuous flow varies if we methodologically prune its neural network architecture. Reducing unnecessary weights of a neural network (pruning) (Han et al., 2015b, Hassibi and Stork, 1993, LeCun et al., 1990, Li et al., 2016) without loss of accuracy results in smaller network size (Hinton et al., 2015, Liebenwein et al., 2021a,b), computational efficiency (Luo et al., 2017, Molchanov et al., 2019, Yang et al., 2017), faster inference (Frankle and Carbin, 2019), and enhanced interpretability (Baykal et al., 2019a,b, Lechner et al., 2020a, Liebenwein et al., 2020). Here, our main objective is to better understand CNFs' dynamics in density estimation tasks as we increase network sparsity and to show that pruning can improve generalization in neural ODEs.

---

[*]denotes authors with equal contributions. Code: https://github.com/lucaslie/torchprune

35th Conference on Neural Information Processing Systems (NeurIPS 2021).

**Pruning improves generalization in neural ODEs.**
Our results consistently suggest that a certain ratio of
pruning of fully connected neural ODEs leads to lower
empirical risk in density estimation tasks, thus obtaining
better generalization. We validate this observation on a
large series of experiments with increasing dimensionality.
See an example here in Figure 2.

**Pruning flattens the loss surface of neural ODEs.** Additionally, we conduct a Hessian-based empirical investigation on the objective function of the flows-under-test in
density estimation tasks to better understand why pruning
results in better generalization. We find that for Neural
ODEs, pruning decreases the value of the Hessian's eigenvalues, and as a result, flattens the loss which leads to
better generalization, c.f., Keskar et al. (2017) (Figure 3).

**Pruning helps avoiding mode-collapse in generative
modeling.** In a series of multi-modal density estimation
tasks, we observe that densely connected CNFs often get
stuck in a sharp local minimum (See Figure 3) and as a
result, cannot properly distinguish different modes of data.
This phenomena is known as mode-collapse. Once we
sparsify the flows, the quality of the density estimation task
increases significantly and consequently mode-collapse
does not occur.

**Pruning finds minimal and efficient neural ODE representations.** Our framework finds highly optimized and
efficient neural ODE architectures via pruning. In many

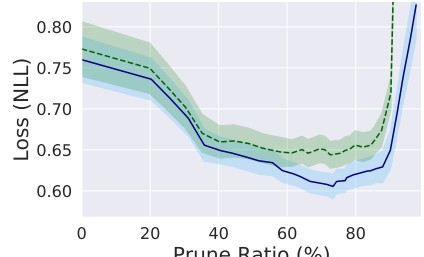

Figure 2: Pruning enhances generalization of continuous-depth models. Structured pruning (green), unstructured pruning (blue). More details in Section 4.

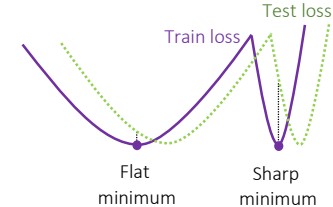

Figure 3: Flat minima result in better generalization compared to sharp minima. Pruning neural ODEs flattens the loss around local minima. Figure is reproduced from Keskar et al. (2017).

instances, we can reduce the parameter count by 70-98% (6x-50x compression rate). Notably, one
cannot directly train such sparse and efficient continuous-depth models from scratch.

## 2   Background

In this section, we describe the necessary background to construct our framework. We show how to
perform generative modeling by continuous depth models using the change of variables formula.

**Generative modeling via change of variables.**   The change of variables formula uses an invertible
mapping $f : \mathbb{R}^D \rightarrow \mathbb{R}^D$, to wrap a normalized base distribution $p_z(\mathbf{z})$, to specify a more complex
distribution. In particular, given $z \sim p_z(\mathbf{z})$, a random variable, the log density for function $f(\mathbf{z}) = \mathbf{x}$
can be computed by Dinh et al. (2015):

$$\log p_x(\mathbf{x}) = \log p_z(\mathbf{z}) - \log \det \left| \frac{\partial f(\mathbf{z})}{\partial \mathbf{z}} \right|, \tag{1}$$

where $\frac{\partial f(\mathbf{z})}{\partial \mathbf{z}}$ is the Jacobian of $f$. While theoretically Eq. 1 demonstrates a simple way to finding
the log density, from a practical standpoint computation of the Jacobian determinant has a time
complexity of $\mathcal{O}(D^3)$. Restricting network architectures can make its computation more tractable.
Examples include designing normalizing flows (Berg et al., 2018, Papamakarios et al., 2017, Rezende
and Mohamed, 2015), autoregressive transformations (Durkan et al., 2019, Jaini et al., 2019, Kingma
et al., 2016, Müller et al., 2019, Oliva et al., 2018a, Wehenkel and Louppe, 2019), partitioned transformations (Dinh et al., 2016, Kingma and Dhariwal, 2018), universal flows (Kong and Chaudhuri,
2020, Teshima et al., 2020), and the use of optimal transport theorem (Huang et al., 2020).

Alternative to these discrete transformation algorithms, one can construct a generative model similar
to (1), and declare $f$ by a continuous-time dynamics (Chen et al., 2018, Grathwohl et al., 2019,
Lechner et al., 2020b). Given a sample from the base distribution, one can parametrize an ordinary
differential equations (ODEs) by a function $f(\mathbf{z}(t), t, \theta)$, and solve the ODE to obtain the observable
data. When $f$ is a neural network, the system is called a neural ODE (Chen et al., 2018).

**Neural ODEs.** More formally, a neural ODE is defined by finding the solution to the initial value problem (IVP): $\frac{\partial \mathbf{z}(t)}{\partial t} = f(\mathbf{z}(t), t, \theta)$, $\mathbf{z}(t_0) = \mathbf{z}_0$, with $\mathbf{z}_0 \sim p_{z_0}(\mathbf{z}_0)$, to get $\mathbf{z}(t_n)$ the desired output observations at a terminal integration step $n$ (Chen et al., 2018).[2]

**Continuous Normalizing Flows.** If $\mathbf{z}(t_n)$ is set to our observable data, given samples from the base distribution $\mathbf{z}_0 \sim p_{z_0}(\mathbf{z}_0)$, the neural ODE described above forms a continuous normalizing flow (CNF). CNFs modify the change in log density by the left hand-side differential equation and as a result the total change in log-density by the right hand-side equation (Chen et al., 2018, Grathwohl et al., 2019):

$$\frac{\partial \log p(\mathbf{z}(t))}{\partial t} = -\text{Tr}\left(\frac{\partial f}{\partial \mathbf{z}(t)}\right), \quad \log p(\mathbf{z}(t_n)) = \log p(\mathbf{z}(t_0)) - \int_{t_0}^{t_1} \text{Tr}\left(\frac{\partial f}{\partial \mathbf{z}(t)}\right) dt. \quad (2)$$

The system of two differential equations (the neural ODE ($\frac{\partial \mathbf{z}(t)}{\partial t} = f(\mathbf{z}(t), t, \theta)$) and (2) can then be solved by automatic differentiation algorithms such as backpropagation through time (Hasani et al., 2021, Rumelhart et al., 1986) or the adjoint sensitivity method (Chen et al., 2018, Pontryagin, 2018). Computation of $\text{Tr}\left(\frac{\partial f}{\partial \mathbf{z}(t)}\right)$ costs $\mathcal{O}(D^2)$. A method called the Free-form Jacobian of Reversible Dynamics (FFJORD) (Grathwohl et al., 2019) improved the cost to $\mathcal{O}(D)$ by using the Hutchinson's trace estimator (Adams et al., 2018, Hutchinson, 1989). Thus, the trace of the Jacobian can be estimated by: $\text{Tr}\left(\frac{\partial f}{\partial \mathbf{z}(t)}\right) = \mathbb{E}_{p(\varepsilon)}\left[\varepsilon^T \frac{\partial f}{\partial \mathbf{z}(t)} \varepsilon\right]$, where $p(\varepsilon)$ is typically set to a Gaussian or Rademacher distribution (Grathwohl et al., 2019). Throughout the paper, we investigate the properties of FFJORD CNFs by pruning their neural network architectures.

## 3  Pruning Neural ODEs

We enable sparsity in neural ODEs and CNFs by removing, i.e., pruning, redundant weights from the underlying neural network architecture during training. Pruning can tremendously improve the parameter efficiency of neural networks across numerous tasks, such as computer vision (Liebenwein et al., 2020) and natural language processing (Maalouf et al., 2021).

### 3.1  A General Framework for Training Sparse Flows

Our approach to training Sparse Flows is inspired by *iterative learning rate rewinding*, a recently proposed and broadly adopted pruning framework as used by Liebenwein et al. (2020), Renda et al. (2020) among others.

In short, our pruning framework proceeds by first training an unpruned, i.e., dense network to obtain a warm initialization for pruning. Subsequently, we proceed by iteratively pruning and retraining the network until we either obtain the desired level of sparsity, i.e., prune ratio or when the loss for a pre-specified hold-out dataset (validation loss) starts to deteriorate (early stopping). We note that our framework is readily applicable to any continuous-depth model and not restricted to FFJORD-like models. Moreover, we can account for various types of pruning, i.e., unstructured pruning of weights and structured pruning of neurons or filters. An overview of the framework is provided in Algorithm 1 and we provide more details below.

### 3.2  From Dense to Sparse Flows

**TRAIN a dense flow for a warm initialization.** To initiate the training process, we first train a densely-connected network to obtain a warm initialization (Line 2 of Algorithm 1). We use Adam with a fixed step learning decay schedule and weight decay in some instances. Based on the warm initialization, we start pruning the network.

**PRUNE for Sparse Flow.** For the prune step (Line 5 of Algorithm 1) we either consider unstructured or structured pruning, i.e., weight or neuron/filter pruning, respectively. At a fundamental level,

---

[2]One can design a more expressive representation (Hasani et al., 2020, Vorbach et al., 2021) of continuous-depth models by using the second-order approximation of the neural ODE formulation (Hasani et al., 2021). This representation might give rise to a better neural flows which will be the focus of our continued effort.

---
**Algorithm 1** SPARSEFLOW($f$, $\Phi_{\text{train}}$, $PR$, $e$)
---
**Input:** $f$: neural ODE model with parameter set $\theta$; $\Phi_{\text{train}}$: hyper-parameters for training; $PR$: relative prune ratio; $e$: number of training epochs per prune-cycle.

**Output:** $f(\cdot; \hat{\theta})$: Sparse Flow; $m$: sparse connection pattern.

1: $\theta_0 \leftarrow$ RANDOMINIT()
2: $\theta \leftarrow$ TRAIN($\theta_0, \Phi_{\text{train}}, e$)   ▷ Initial training stage with dense neural ODE ("warm start").
3: $m \leftarrow 1^{|\theta_0|}$   ▷ Initialize binary mask indicating neural connection pattern.
4: **while** validation loss of Sparse Flow decreases **do**
5:    $m \leftarrow$ PRUNE($m \odot \theta, PR$)   ▷ Prune $PR\%$ of the *remaining* parameters and update mask.
6:    $\theta \leftarrow$ TRAIN($m \odot \theta, \Phi_{\text{train}}, e$)   ▷ Restart training with updated connection pattern.
7: **end while**
8: $\hat{\theta} \leftarrow m \odot \theta$,    and    **return** $f(\cdot; \hat{\theta}), m$
---

unstructured pruning aims at inducing sparsity into the parameters of the flow while structured pruning enables reducing the dimensionality of each flow layer.

For unstructured pruning, we use magnitude pruning (Han et al., 2015a)), where we prune weights across all layers (global) with magnitudes below a pre-defined threshold. For structured pruning, we use the $\ell_1$-norm of the weights associated with the neuron/filter and prune the structures with lowest norm for constant per-layer prune ratio (local) as proposed by Li et al. (2016). See Table 1 for an overview.

Table 1: Pruning Methods.

|  | Unstructured (Han et al., 2015a) | Structured (Li et al., 2016) |
|---|---|---|
| **Target** | Weights | Neurons |
| **Score** | $\|W_{ij}\|$ | $\|W_{i:}\|_1$ |
| **Scope** | Global | Local |

**TRAIN the Sparse Flow.**    Following the pruning step, we re-initiate the training with the new sparsity pattern and the unpruned weights (Line 6 of Algorithm 1). Note that we do not change the training hyperparameters between the different stages of training.

**ITERATE for increased sparsity and performance.**    Naturally, we can iteratively repeat the PRUNE and TRAIN step to further sparsify the flow (Lines 4-7 of Algorithm 1). Moreover, the resulting sparsity-performance trade-off is affected by the total number of iterations, the relative prune ratio $PR$ per iteration, and the amount of training between PRUNE steps. We find that a good trade-off is to keep the amount of training constant across all iterations and tune it such that the initial, dense flow is essentially trained to (or close to) convergence. Depending on the difficulty of the task and the available compute resources we can then adapt the per-iteration prune ratio $PR$. Note that the overall relative sparsity after $n$ iterations is given by $(1 - PR)^n$. Detailed hyperparameters and more explanations for each experiment are provided in the supplementary material.

## 4   Experiments

We perform a diverse set of experiments demonstrating the effect of pruning on the generalization capability of continuous-depth models. Our experiments include pruning ODE-based flows in density estimation tasks with increasing complexity, as well as pruning neural-ODEs in supervised inference tasks. The density estimation tasks were conducted on flows equipped with Free-form Jacobian of Reversible Dynamics (FFJORD) (Grathwohl et al., 2019), using adaptive ODE solvers (Dormand and Prince, 1980). We used two code bases (FFJORD from Grathwohl et al. (2019) and TorchDyn (Poli et al., 2020a)) over which we implemented our pruning framework.[3]

**Baselines.** In complex density estimation tasks we compare the performance of Sparse Flows to a variety of baseline methods including: FFJORD (Grathwohl et al., 2019), masked autoencoder density estimation (MADE) (Germain et al., 2015), Real NVP (Dinh et al., 2016), masked autoregressive flow (MAF) (Papamakarios et al., 2017), Glow (Kingma and Dhariwal, 2018), convex potential flows (CP-Flow) (Huang et al., 2020), transformation autoregressive networks (TAN) (Oliva et al., 2018b), neural autoregressive flows (NAF) (Huang et al., 2018), and sum-of-squares polynomial flow (SOS) (Jaini et al., 2019).

---

[3]All code and data are available online at: `https://github.com/lucaslie/torchprune`

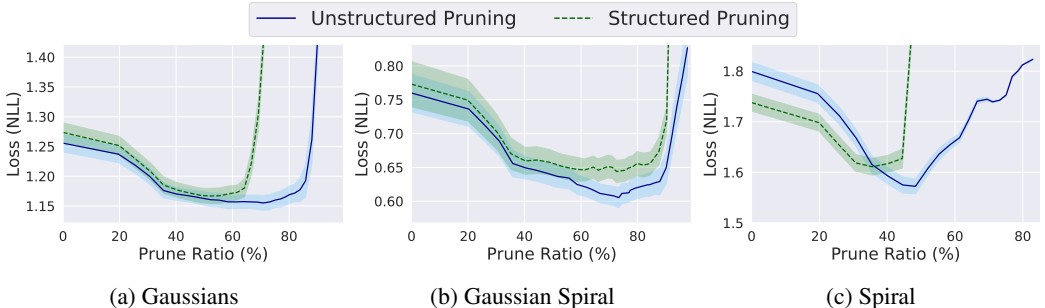

(a) Gaussians          (b) Gaussian Spiral          (c) Spiral

Figure 4: Negative log likelihood of Sparse Flow as function of prune ratio. In general, structured pruning is a more constrained problem as we constrain the type of sparsity. Hence, for almost any pruning experiment we can expect that structured pruning performs worse than unstructured pruning starting at a certain prune ratio.

## 4.1 Density Estimation on 2D Data

In the first set of experiments, we train FFJORD on a multi-modal Gaussian distribution, a multi-model set of Gaussian distributions placed orderly on a spiral as well as a spiral distribution with sparse regions. Figure 5 (first row) illustrates that densely connected flows (prune ratio = 0%) might get stuck in sharp local minima and as a result induce mode collapse (Srivastava et al., 2017a). Once we perform unstructured pruning, we observe that the quality of the density estimation in all tasks considerably improves, c.f. Figure 5 (second and third rows). If we continue sparsifying the flows, depending on the task at hand, the flows get disrupted again.

Therefore, there is a certain threshold for pruning flows required to avoid generative modeling issues such as mode-collapse in continuous flows. We validate this observation by plotting the negative log-likelihood loss as a function of the prune ratio in all three tasks with both unstructured and structured pruning. As shown in Figure 4, we confirm that sparsity in flows improves the performance of continuous normalizing flows.

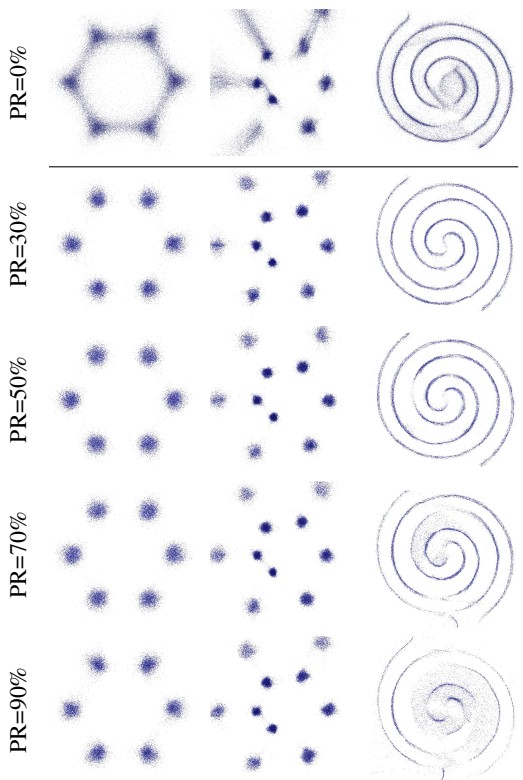

Figure 5: Pruning FFJORD (PR= Prune ratio).

We further explore the inner dynamics of the flows between unpruned and pruned networks on the multi-modal case, with the aim of understanding how pruning enhances density estimation performance. Figure 6 represents the vector-field constructed by each flow to model 6-Gaussians independently. We observe that sparse flows with PR of 70% attract the vector-field directions uniformly towards the mean of each Gaussian. In contrast, unpruned flows do not exploit this feature and contain converging vectors in between distributions. This is how the mode-collapse occurs.

## 4.2 Density Estimation on Real Data - Tabular

We scale our experiments to a set of five real-world tabular datasets (prepared based on the instructions given by Papamakarios et al. (2017) and Grathwohl et al. (2019)) to verify our empirical observations about the effect of pruning on the generalizability of continuous normalizing flows. Table 2 summarizes the results. We observe that sparsifying FFJORD flows substantially improves

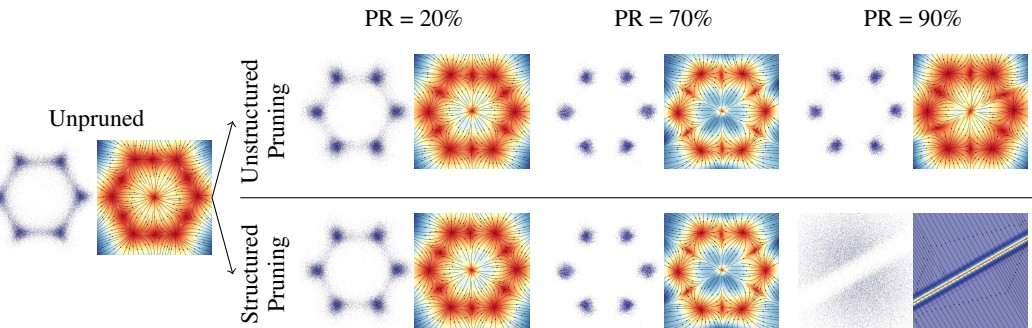

Figure 6: Multi-modal Gaussian flow and pruning. We observe that Sparse Flows attract the vector-field directions uniformly towards the mean of each Gaussian distribution, while an unpruned flow does not exploit this feature and contains converging vectors in between Gaussians. See Supplements Section S2, for a detailed explanation of these observations.

Table 2: Negative test log-likelihood (NLL) in nats of tabular datasets from (Papamakarios et al., 2017) and corresponding architecture size in number of parameters (#params). Sparse Flow (FFJORD with unstructured pruning) with lowest NLL and competing baseline with lowest NLL are bolded.

| Model | POWER | | GAS | | HEPMASS | | MINIBOONE | | BSDS300 | |
|---|---|---|---|---|---|---|---|---|---|---|
| | nats | #params | nats | #params | nats | #params | nats | #params | nats | #params |
| MADE (Germain et al., 2015) | 3.08 | 6K | -3.56 | 6K | 20.98 | 147K | 15.59 | 164K | -148.85 | 621K |
| Real NVP (Dinh et al., 2016) | -0.17 | 212K | -8.33 | 216K | 18.71 | 5.46M | 13.84 | 5.68M | -153.28 | 22.3M |
| MAF (Papamakarios et al., 2017) | -0.24 | 59.0K | -10.08 | 62.0K | 17.70 | 1.47M | 11.75 | 1.64M | -155.69 | 6.21M |
| Glow (Kingma and Dhariwal, 2018) | -0.17 | N/A | -8.15 | N/A | 18.92 | N/A | 11.35 | N/A | -155.07 | N/A |
| CP-Flow (Huang et al., 2020) | -0.52 | 5.46M | -10.36 | 2.76M | 16.93 | 2.92M | 10.58 | 379K | -154.99 | 2.15M |
| TAN (Oliva et al., 2018b) | **-0.60** | N/A | **-12.06** | N/A | **13.78** | N/A | 11.01 | N/A | **-159.80** | N/A |
| NAF (Huang et al., 2018) | **-0.62** | 451K | **-11.96** | 443K | 15.09 | 10.7M | **8.86** | **8.03M** | -157.73 | 42.3M |
| SOS (Jaini et al., 2019) | **-0.60** | 212K | **-11.99** | 256K | 15.15 | 4.43M | **8.90** | **6.87M** | -157.48 | 9.09M |
| FFJORD (Grathwohl et al., 2019) | -0.35 | 43.3K | -8.58 | 279K | 17.53 | 547K | 10.50 | 821K | -128.33 | 6.70M |
| Sparse Flow | -0.45 | 30K | -10.79 | 194K | 16.53 | 340K | 10.84 | 397K | -145.62 | 4.69M |
| | -0.50 | 23K | -11.19 | 147K | 15.82 | 160K | 10.81 | 186K | -148.72 | 3.55M |
| | **-0.53** | **13K** | **-11.59** | **85K** | **15.60** | **75K** | **9.95** | **32K** | -150.45 | 2.03M |
| | -0.52 | 10K | -11.47 | 64K | 15.99 | 46K | 10.54 | 18K | **-151.34** | **1.16M** |

their performance in all 5 tasks. In particular, we gain up to 42% performance gain in the POWER, 35% in GAS, 12% in HEPMASS, 5% in MINIBOONE and 19% in BSDS300.

More importantly, this is achieved with flows with 1 to 3 orders of magnitude less parameters compared to other advanced flows. On MINIBOONE dataset for instance, we found a sparse flow with only 4% of its original network that outperforms its densely-connected FFJORD flow. On MINIBOONE, Autoregressive flows (NAF) and sum-of-squares models (SOS) which outperform all other models possess 8.03 and 6.87 million parameters. In contrast, we obtain a Sparse Flow with only 32K parameters that outperform all models except NAF and SOS.

Let us now look at the loss versus prune-ratio trends in all experiments to conclude our empirical observations on real-world tabular datasets. As shown in Figure 7, we observe that pruning considerably improves the performance of flows at larger scale as well.

## 4.3 Density Estimation on Real Data - Vision

Next, we extend our experiments to density estimation for image datasets, MNIST and CIFAR10. We observe a similar case on generative modeling with both datasets, where pruned flows outperform densely-connected FFJORD flows. On MNIST, a sparse FFJORD flow with 63% of its weights pruned outperforms all other benchmarks. Compared to the second best flow (Residual flow), our

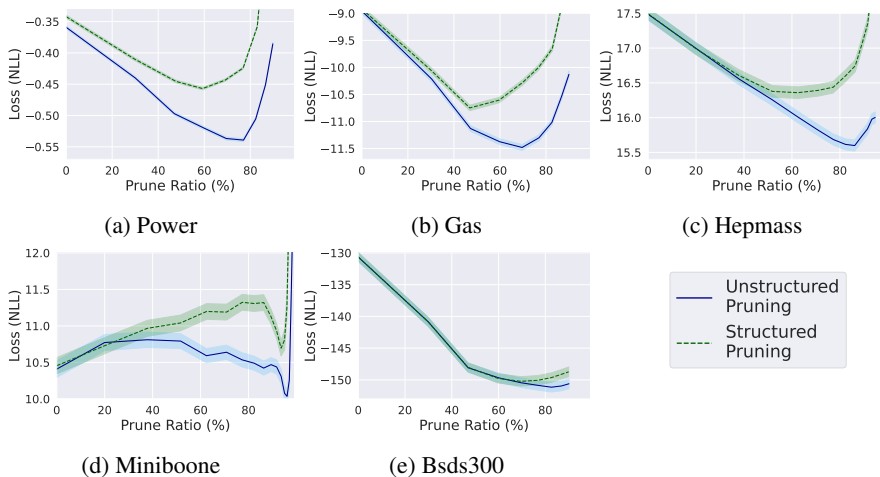

(a) Power  (b) Gas  (c) Hepmass

(d) Miniboone  (e) Bsds300

Figure 7: Negative log-likelihood versus prune ratio on tabular datasets.

Table 3: Negative test log-likelihood (NLL) in bits/dim for image datasets and corresponding architecture size in number of parameters (#params). Sparse Flow (FFJORD with unstructured pruning) with lowest NLL and competing baseline with lowest NLL are bolded.

| Model | MNIST | | CIFAR-10 | |
| --- | --- | --- | --- | --- |
| | bits/dim | #params | bits/dim | #params |
| MADE (Germain et al., 2015) | 1.41 | 1.20M | 5.80 | 11.5M |
| Real NVP (Dinh et al., 2016) | 1.05 | N/A | 3.49 | N/A |
| MAF (Papamakarios et al., 2017) | 1.91 | 12.0M | 4.31 | 115M |
| Glow (Kingma and Dhariwal, 2018) | 1.06 | N/A | 3.35 | 44.0M |
| CP-Flow (Huang et al., 2020) | 1.02 | 2.90M | 3.40 | 1.90M |
| TAN (Oliva et al., 2018b) | 1.19 | N/A | 3.98 | N/A |
| SOS (Jaini et al., 2019) | 1.81 | 17.2M | 4.18 | 67.1M |
| RQ-NSF (Durkan et al., 2019) | | N/A | 3.38 | 11.8M |
| Residual Flow (Chen et al., 2019) | **0.97** | **16.6M** | **3.28** | **25.2M** |
| FFJORD (Grathwohl et al., 2019)) | 1.01 | 801K | 3.44 | 1.36M |
| Sparse Flows (PR=20%) | 0.97 | 641K | 3.38 | 1.09M |
| Sparse Flows (PR=38%) | 0.96 | 499K | 3.37 | 845K |
| Sparse Flows (PR=52%) | 0.95 | 387K | **3.36** | **657K** |
| Sparse Flows (PR=63%) | **0.95** | **302K** | 3.37 | 510K |
| Sparse Flows (PR=71%) | 0.96 | 234K | 3.38 | 395K |
| Sparse Flows (PR=77%) | 0.97 | 182K | 3.39 | 308K |
| Sparse Flows (PR=82%) | 0.98 | 141K | 3.40 | 239K |
| Sparse Flows (PR=86%) | 0.97 | 109K | 3.42 | 186K |

sparse flow contains 70x less parameters (234K vs 16.6M). On CIFAR10, we achieve the second best performance with over 38x less parameters compared to Residual flows which performs best.

Furthermore, on CIFAR10, Glow with 44 million parameters performs on par with our sparse FFJORD network with 657k parameters. It is worth mentioning that FFJORD obtains this results by using a simple Gaussian prior, while Glow takes advantage of learned base distribution (Grathwohl et al., 2019). Figure 8 illustrates the improvement of the loss (negative log-likelihood) in density estimation on image datasets as a result of pruning neural

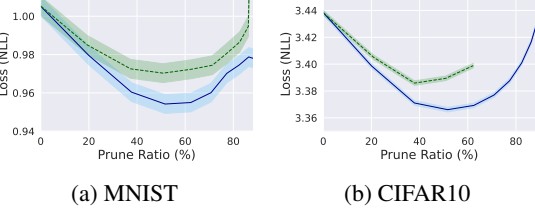

(a) MNIST  (b) CIFAR10

Figure 8: Loss vs. prune ratio for MNIST and CIFAR10 with unstructured and structured pruning.

Table 4: Percentage of good quality samples as a measure of mode-collapse. Pruning method: structured pruning.

| | GAUSSIAN | | GAUSSIAN-SPIRAL | | MNIST | | |
|---|---|---|---|---|---|---|---|
| | Percentage of good quality samples as a measure of mode-collapse | | | | | | |
| Prune ratio (%) | std = 2 | std = 3 | std = 2 | std = 3 | Prune ratio (%) | std = 3 | std = 5 |
| 0.0 | 72.89% | 89.34% | 66.65% | 84.01% | 0.0 | 6.94% | 58.24% |
| 20.0 | 79.46% | 93.45% | 74.29% | 90.64% | 20.0 | 8.02% | 65.98% |
| **25.6** | **81.40%** | **94.90%** | **78.22%** | **93.02%** | **37.8** | **9.35%** | **68.11%** |
| **52.0** | **84.30%** | **96.85%** | **80.40%** | **94.29%** | **51.7** | **8.70%** | **67.24%** |
| **71.2** | **81.83%** | **94.86%** | **78.06%** | **93.41%** | **77.3** | **8.59%** | **66.24%** |
| 75.1 | 68.20% | 86.10% | 78.99% | 93.67% | 89.3 | 6.07% | 57.28% |
| 98.0 | 36.01% | 76.54% | 9.27% | 20.32% | 91.7 | 0.02% | 32.75% |

ODEs. Around 60% sparsity of a continuous normalizing flow leads to better generative modeling compared to densely structured flows.

## 4.4 Pruning Helps Avoid Mode-collapse

To quantitatively explore the effectiveness of pruning on avoiding mode-collapse (as shown qualitatively in Figure 5, we developed a measure adopted from (Srivastava et al., 2017b): The percentage of Good Quality samples— We draw samples from a trained normalizing flow and consider a sample of "good quality" if it is within $n$ ($n$ is typically chosen to be 2, 3, or 5) standard deviations from its nearest mode. We then can report the percentage of the good quality samples as a measure of how well the generative model captures modes.

Table 4 summarizes the results on some synthetic datasets and the large-scale MNIST dataset. We can see that the number of "good quality samples", which stand for a quantitative metric for mode-collapse, improves consistently with a certain percentage of pruning in all experiments. This observation is very much aligned with our results on the generalization loss, which we will discuss in the following.

## 4.5 Pruning Flattens the Loss Surface for Neural ODEs

What could be the potential reason for the enhanced generalization of pruned CNFs besides their ability to resolve mode-collapse? To investigate this further, we conducted a Hessian-based analysis on the flows-under-test. Dissecting the properties of the Hessian by Eigenanalysis allows us to gain useful insights about the behavior of neural networks (Erichson et al., 2021, Ghorbani et al., 2019, Hochreiter and Schmidhuber, 1997, Lechner and Hasani, 2020, Sagun et al., 2017). We use PyHessian (Yao et al., 2020) tool-set to analyze the Hessian $H$ w.r.t. the parameters of the CNF. This enables us to study the curvature of the loss function as the eigenvalues of the Hessian determines (locally) the loss gradients' changes.

Table 5: Eigenanalysis of the Hessian $H$ in terms of the largest eigenvalue ($\lambda_{max}$), trace (tr), and condition number ($\kappa$) of pruned and unpruned continuous normalizing flows on the mixture of Gaussian task. Numbers are normalized with respect to the unpruned flow. See more results on other datasets in the supplements Section S3.

| Model | NLL | $\lambda_{max}(H)$ | $\text{tr}(H)$ | $\kappa(H)$ |
|---|---|---|---|---|
| Unpruned FFJORD | 1.309 | 1.000 | 1.000 | 1.000 |
| Sparse Flows (PR=20%) | 1.163 | 0.976 | 0.858 | 0.825 |
| Sparse Flows (PR=60%) | 1.125 | 0.356 | 0.583 | 0.717 |
| Sparse Flows (PR=70%) | **1.118** | **0.295** | **0.340** | **0.709** |
| Sparse Flows(PR=90%) | 1.148 | 0.416 | 0.366 | 0.547 |

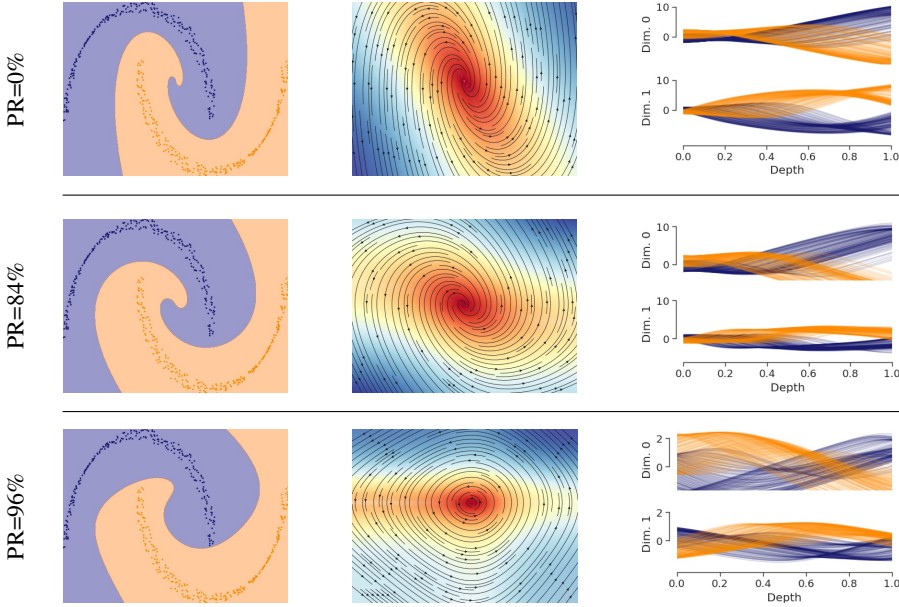

Figure 9: Robustness of decision boundaries for pruned networks. Column 1 is the decision boundary. Column 2 = state-space, and column 3 = the flow of data points: it shows the evolution of single samples (each narrow line represents an individual sample's dimension 1 path) from input to their output through the depth of the network. How does the attenuation of the vector field intensity indicate that the classification boundary is more sensitive to perturbations? Let's look at the first column and row 3 entry: We see that the classification boundary between the blue half-moon and the orange half-moon is very close to one tail of the blue half-moon. This means that if we perturb the blue half-moon data, some of them might get classified incorrectly based on the extremely pruned network's decision boundary. As the intensity of the vector field shrinks to the center of the vector field (it is attenuated in the areas distant from the center), the decision boundary is set poorly as opposed to networks with more parameters.

Larger Hessian eigenvalues therefore, stand for sharper curvatures and their sign identifies upward or downward curvatures. In Table 5, we report the maximum eigenvalue of the Hessian $\lambda_{max}$, Hessian's Trace, and the condition number $\kappa = \frac{\lambda_{max}}{\lambda_{min}}$.[4] Smaller $\lambda_{max}$ and tr($H$) indicates that our normalizing flow found a flatter minimum. As shown in Figure 3, a flat minimum leads to a better generalization error as opposed to a sharp minimum. We find that up to a certain prune ratio, the maximum of the Hessian decreases and increases thereafter. Therefore, we claim that pruned continuous flows finds flatter local minimum, therefore, it generalize better than their unpruned version.

Moreover, the Hessian condition number $\kappa$ could be an indicator of the robustness and efficiency of a deep model (Bottou and Bousquet, 2008). Smaller $\kappa$ corresponds to obtaining a more robust learned agent. We observe that $\kappa$ also follows the same trend and up to a certain prune ratio it shrinks and then increases again confirming our hypothesis.

## 4.6 On the Robustness of Decision Boundaries of Pruned ODE-based Flows

While pruning improves performance, it is imperative to investigate their robustness properties in constructing decision boundaries (Lechner et al., 2021). For feedforward deep models it was recently shown that although pruned networks perform as well as their unpruned version, their robustness significantly decreases due to smaller network size (Liebenwein et al., 2021a). Is this also the case for neural ODEs?

We design an experiment to investigate this. We take a simple 2-Dimensional moon dataset and perform classification using unpruned and pruned neural ODEs.[5] This experiment (shown in Figure

---

[4]This experiment was inspired by the Hessian-based robustness analysis performed in Erichson et al. (2021).
[5]This experiment is performed using the TorchDyn library (Poli et al., 2020a).

9) demonstrates another valuable property of neural ODEs: We observe that neural ODE instances pruned up to 84%, manage to establish a safe decision boundary for the two half moon classes. This insight about neural ODEs which was obtained by our pruning framework is another testimony of the merits of using neural ODE in decision-critical applications. Additionally, we observe that the decision-boundary gets very close to one of the classes for a network pruned up to 96% which reduces robustness. Though even this network instance provide the same classification accuracy compared to the densely connected version.

The state-space plot illustrates that a neural ODE's learned vector-field becomes edgier as we keep pruning the network. Nonetheless, we observe that the distribution of the vector-field's intensity (the color map in the second column) get attenuated as we go further away from its center, in highly pruned networks. This indicates that classification near the decision boundary is more sensitive to perturbations in extremely pruned networks.

Moreover, Column 3 of Figure 9 shows that when the networks are not pruned (column 3, row 1) or are pruned up to a certain threshold (column 3, row 2), the separation between the flow of samples of each class (orange or blue) is more consistent compared to the very pruned network (column 3, row 3). In Column 3, row 3 we see that the flow of individual samples from each class is more dispersed. As a result, the model is more sensitive to perturbations on the input space. This is very much aligned with our observation from column 1 and column 2 of Figure 9.

# 5 Discussions, Scope and Conclusions

We showed the effectiveness of pruning for continuous neural networks. Pruning improved generalization performance of continuous normalizing flows in density estimation tasks at scale. Additionally, pruning allows us to obtain performant minimal network instances with at least one order of magnitude less parameter count. By providing key insights about how pruning improves generative modeling and inference, we enabled the design of better neural ODE instances.

**Ensuring sparse Hessian computation for sparse continuous-depth models.** As we prune neural ODE instances, their weight matrices will contain zero entries. However the Hessian with respect to those zero entries is not necessarily zero. Therefore, when we compute the eigenvalues of the Hessian, we must ensure to make the decompositon vector sets the Hessian of the pruned weights to zero before performing our eigenanalysis.

**Why experimenting with FFJORD and not CNFs?** FFJORD is an elegant trick to efficiently compute the log determinant of the Jacobians in the change of variables formula for CNFs. Thus, FFJORDs are CNFs but faster. To confirm the effectiveness and scalability of our approach, we pruned some CNFs on toy datasets and concluded very similar outcomes (see these results in Appendix).

**What are the limitations of Sparse Flows?** Similar to any ODE-based learning system, the computational efficiency of sparse flows is highly determined by the choice of their ODE solvers, data and model parameters. As the complexity of any of these fronts increases, the number of function evaluations for a given task increases. Thus we might have a slow training process. This computational overhead can be relaxed in principle by the use of efficient ODE solvers (Poli et al., 2020b) and flow regularization schemes (Massaroli et al., 2020).

**What design notes did we learn from applying pruning to neural ODEs?** We performed an ablation study over different types of features of neural ODE models. These experiments can be found in the supplements Section S4. Our framework suggested that to obtain a generalizable sparse neural ODE representation, the choice of activation function is important. In particular activations that are Lipschitz continuous, monotonous, and bounded are better design choices for density estimation tasks.

Moreover, we found that the generalizability of sparse neural ODEs is more influenced by their neural network's width than their depth (number of layers), c.f., Appendix Section S4. Furthermore, pruning neural ODEs allows us to obtain better hyperparameters for the optimization problem by setting a trade-off between the value of the learning rate and weight decay.

In summary, we hope to have shown compelling evidence for the effectiveness of having sparsity in ODE-based flows.

## Acknowledgments

This research was sponsored by the United States Air Force Research Laboratory and the United States Air Force Artificial Intelligence Accelerator and was accomplished under Cooperative Agreement Number FA8750-19-2-1000. The views and conclusions contained in this document are those of the authors and should not be interpreted as representing the official policies, either expressed or implied, of the United States Air Force or the U.S. Government. The U.S. Government is authorized to reproduce and distribute reprints for Government purposes notwithstanding any copyright notation herein. This work was further supported by The Boeing Company and the Office of Naval Research (ONR) Grant N00014-18-1-2830.

## Funding Transparency Statement

Authors declare no competing interests. *Funding in direct support of this work:* The Boeing Company through the Explainable Control Project, National Science Foundation (NSF) Graduate Research Fellowship Program, the United States Air Force Research Laboratory and the United States Air Force Artificial Intelligence Accelerator and was accomplished under Cooperative Agreement Number FA8750-19-2-1000, and The Boeing Company and the Office of Naval Research (ONR) Grant N00014-18-1-2830.

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
