# OpenReview forum: "Sparse Flows: Pruning Continuous-depth Models"
_NeurIPS.cc/2021/Conference — NeurIPS 2021 Poster_

### Official Review · Reviewer_AFJg · 2021-07-15

**Rating:** 6
**Confidence:** 3

**Summary:**

The authors introduce sparse-flows, which is trained by iteratitively pruning a continous normalizing flow network (in this case FFJORD).

Through their experiments the authors show that the sparsified FFJORD model often preforms better than FFJORD and other flow-based models like GLOW, real-NVP etc. They show (on a variety of datasets) that upto a limit, sparsifiying the model can reduce the NLL of the data. The authors also show that pruning flattens the loss surface for Neural-ODEs and show that classifiers trained through sparse-ODEs are more robust than ODEs.

**Limitations And Societal Impact:**

The authors have adequately discussed the limitations of their work.

**Main Review:**

The paper is well written and motivated and the experiments provide good insights on the effects of sparsifying continuous normalizaing flows.

While the results are good, my concern is with the overall novelty of the paper. The authors tested out the results of pruning in a new field, namely continuous flow, however their findings although intriguing are not so suprising given the recent results papers that show that neural networks architectures are sparse and thus can be pruned to reduce their size while keeping the performace intact.

Sparse Flow, the algorithm introduced by the authors, is based on FFJORD (which I presume is more efficient to train than other models). However, given that the main contribution of the paper is how sparsity improves the performance of flow-based models, the methodology should be applied to other models as well, especially to verify the performance gain is not only limited to sparsifying FFJORD.

**Time Spent Reviewing:**

3.5

---

> ### Author Response · Authors · 2021-08-11
> **Response to Reviewer AFJg**
>
> We thank the reviewer for commenting on our paper. We would like to emphasize that improving the generalization of continuous flows is not a trivial task that is enabled by our pruning framework. Moreover finding out “why and how pruning improves generalization in continuous flows” is what we showed in our paper (Please see the discussions about prevention of model collapse, sharp local minima vs flat local minima, the Hessian-based sensitivity analysis, our large-scale experiments in generative modeling, and our discussions on the confidence bounds of a continuous network classifier).
>
> Additionally, FFJORD is an elegant estimation technique applied to the general class of CNFs for improving the speed of a CNF (by efficiently estimating the log determinant of the Jacobians in the change of variable formula). We focused our experiments on CNFs supplied with FFJORD to speed up our experiments. As we showed in an additional experiment (as a response to Reviewer LH4D), we confirmed that our observations extend consistently well to CNFs regardless of how efficiently we compute their log determinant of the Jacobians. Sparsifying discrete flows is an interesting idea to be tested which was not the focus of our current research.
>
> Finally, please denote that our findings suggest a significant improvement in the number of parameters vs loss trade-offs for density estimation tasks for continuous flows, with much quantitative evidence on how and why this trade-off exists. This is of significant impact and interest for the Neural ODE community as our work sheds more light on designing better ODE-based neural systems.
>
> We hope to have addressed the reviewer’s concerns about our work’s impact and novelty and to have encouraged their evaluation towards a more positive assessment, similar to other positive reviews.  We would be more than happy to address any remaining concerns the reviewer might have on our manuscript.

---

> > ### Comment · Reviewer_AFJg · 2021-08-23
> > **Reply**
> >
> > Thank you for your clarification and the additional results. The results on CNFs and also the mode collapse experiments make this a stronger contribution than before. Thank you.

---

### Official Review · Reviewer_LH4D · 2021-07-15

**Rating:** 7
**Confidence:** 3

**Summary:**

This paper shows that it is possible to drastically prune the network parameterising the neural ODE in a CNF and that this can result in improved generalisation as measured by the LL on a held-out test dataset. The authors hypothesise that this is due to the flattening of the loss surface around the local minima.

**Limitations And Societal Impact:**

The authors discussed the limitations of their work but did not touch on any societal impacts.

**Main Review:**

While the idea of this paper is not particularly original – the authors apply existing pruning techniques to an existing CNF method – the simplicity is a plus in this case. A simple paper is in a prime position to have a big practical impact, and smaller neural networks, which are *also more performant*, are of course going to be of great interest to ML practitioners. However, the paper's potentially large practical significance is, unfortunately, let down by a few half-baked ideas, some overly strong claims, and lack of clarity in a few places.

## Half-baked ideas & overly strong claims

1. *Structured vs unstructured pruning.* The authors mention that their method can be applied to both kinds of pruning, and shows that this is true for one of their toy experiments. However, they don't demonstrate that this holds for larger-scale experiments. Nor do they provide any analysis comparing/contrasting these methods in the context of CNF pruning which is a shame. These kinds of insights would be very interesting and useful!

2. *Key insights enabling the design of better neural ODE instances.* It wasn't clear to me what the insights were that would allow for the design of better neural ODEs. Furthermore, I would want to see some proof-of-concept experiments that show that this is true. The authors might be referring to the paragraph "What design notes did we learn from applying pruning to neural ODEs?" but most of these insights don't seem to be related to pruning. Furthermore, none of the claims here were demonstrated in the paper. Experiments should be provided to back these claims up, at the moment it isn't clear where they come from.

3. *Pruning helps to avoid mode-collapse in generative modelling.* This is a strong claim to make given that this is only demonstrated on a few toy datasets.

4. *Pruning Flattens the Loss Surface for Neural ODEs.* This is also a strong claim to make, given evaluation on a single dataset – results for more datasets (including tabular and image) would be helpful.  Relatedly, "We find that up to a certain prune ratio, the maximum of the Hessian decreases and increases thereafter" is also not shown in the results. Specifically, no increase is shown, and only a few prune ratios are shown – a plot with a range of ratios would be much more illustrative.

5. *One cannot directly train sparse and efficient CNFs from scratch.* While this sounds like a reasonable statement, it is not necessarily true, a small experiment to demonstrate this would help! For example, in tables 2 & 3, add results for some FFJORD models with the same number of parameters as the pruned versions.

6. It would be great to have a few experiments showing whether or not the results from this paper do indeed also apply to standard CNFs (without FFJORD) or if there is something special about FFJORD which makes pruning work well.

## Clarity issues/questions

1. It isn't clear where the 98% pruning mentioned in the abstract comes from.

2. It isn't clear what the term "decipher the internal dynamics" in the abstract means.

3. Please be more specific about the pruning procedure. From reading the text I am not sure I'd be able to replicate the results. Saying that the procedure is "inspired" by iterative learning rate rewinding is too vague! Similarly, the term "pre-defined threshold" isn't specific enough, it isn't clear exactly what is meant by "prune the structures" (what kinds of structures are being considered?), and it isn't clear what "we generally follow the same hyperparameters" means?

4. I did not understand figure 6. A more detailed description, some annotations in the figure, and perhaps adding another dataset would help to know what I should be seeing.

5. What are the pruning ratios in table 2, and how were they chosen? Why do some datasets have more pruning variations than others?

6. Why do figures 8 a), b), and e) not have the full range of pruning ratios?

7. What does the phrase "flows-under-test" mean?

8. Regarding figure 9:
  * What does it mean for a vector field to be "edgier"? Annotating the figure might help here.
  * How does the attenuation of the vector field intensity indicate that the classification boundary is more sensitive to perturbations? A little intuition might be helpful here.
  * What should I be looking at in column 3 of figure 9? Some analysis would be helpful.

## Other comments & suggestions

1. I think the NICE paper would be a better citation for eqn 1.

2. Add NSF as a benchmark for the tabular experiments.

3. Split eqn 2 into two equations, and add a numbered equation for the neural ODE formula on line 100. Then it would be much easier/clearer to reference these equations in the paragraph "Continuous Normalizing Flows".

4. Add a citation for the use of back-prop through time to train CNFs?

5. In figure 5, it would be very helpful to show the ground truth densities. Otherwise, it is hard to know how the method is performing.

## Summary

Overall, I think this paper has a lot of potential to be both interesting and useful. However, I think the current iteration falls short of this promise. As a result, I will be giving a score of 4. I would be very happy to increase my score if the authors can expand on what they have already and provide a more comprehensive and clear story of pruning in CNFs.

## Update during the discussion phase

The authors have addressed a number of my concerns and clarified many sources of confusion. I do still have a number of questions that I've discussed in responses to the authors. However, I am happy to increase my score to a 5 given the response so far and would be willing to increase it further depending on the response to my remaining concerns.

## Final update during the discussion phase

The authors have addressed my remaining concerns and have added substantial additional experimental results. As a result, I have decided to increase my score to a 7. I have, however, decreased my confidence as it is hard to know exactly what the camera-ready version of the paper will now look like.

**Time Spent Reviewing:**

7

---

> ### Author Response · Authors · 2021-08-11
> **Response to Reviewer LH4D - PART 1 of 2**
>
> We thank the reviewer for their detailed review of our manuscript. We very much appreciate the openness of the reviewer to increase their score given that the issues raised will be addressed. Please find our response here:
>
> ### Comments on ideas and results presented in the paper
>
> **Comparing structured and unstructured pruning:**
> We thank the reviewer for this valuable suggestion. We agree that these experiments will provide impactful insights. As per the reviewer's request, we initiated experiments for structured pruning on MINIBOONE, HEPMASS, and MNIST. Our preliminary results are given below:
>
> MINIBOONE (WT from Table 2, FT from new experiments):
>
> | SparseFlows model | nats | # params |
> | --- | --- | --- |
> | Unpruned (FFJORD) | 10.50 | 821k|
> | WT, PR=52% | 10.84 | 397k |
> | FT, PR=52% | 11.37 | 397k |
> | WT, PR=78% | 10.81 | 186k |
> | FT, PR=78% | 10.35 | 186k |
> | WT, PR=96% | 9.95 | 32k |
> | FT, PR=96% | 11.21 | 32k |
> | WT, PR=98% | 10.54 | 18k |
> | FT, PR=98% | 14.74 | 18k |
>
> HEPMASS (WT from Table 2, FT from new experiments):
>
> | SparseFlows model | nats | # params |
> | --- | --- | ---|
> | Unpruned (FFJORD) | 17.53 | 547k |
> | WT, PR=38% | 16.53 | 340k |
> | FT, PR=38% | 16.52 | 340k |
> | FT, PR=62% | 16.21 | 206k |
> | WT, PR=71% | 15.82 | 160k |
>
> MNIST (WT from Table 3, FT from new experiments):
>
> | SparseFlows model | bits/dim | # params |
> | --- | --- | ---|
> | Unpruned (FFJORD) | 1.01 | 801k |
> | WT, PR=20% | 0.97 | 641k |
> | FT, PR=20% | 0.98 | 641k |
> | WT, PR=38% | 0.96 | 499k |
> | FT, PR=38% | 0.97 | 499k |
> | WT, PR=52% | 0.95 | 387k |
> | FT, PR=52% | 0.97 | 387k |
> | WT, PR=63% | 0.95 | 302k |
> | FT, PR=63% | 0.97 | 302k |
>
> From our preliminary results, we may conclude that our main observation about structured vs. unstructured pruning readily scales to larger datasets as well:
>
> _Structured pruning induces a similar regularization effect as unstructured pruning to improve the generalization performance of flows. However, the “best performing” prune ratio and resulting test loss might be slightly worse than the corresponding result for unstructured pruning as structured pruning is a more constrained procedure._
>
> We will continue to experiment with more prune ratios and expand our analysis of structured pruning to the remaining datasets as well (Power, Gas, Bsds300, CIFAR10). As results become available, we will continue to update our response to reflect the latest results.
>
> **Key insights for designing better Neural ODE instances:**
> The first and foremost insight is *the application of pruning itself to an ODE-based flow*. If we supply a continuous flow with pruning, we show that regardless of the size of the dataset, we gain better generalizability. These results are extensively shown throughout our experimentation on toy examples, real-world datasets, and large-scale image datasets. Secondly, please see our detailed response to Reviewer uTzg about lines 299-306.
>
> **Avoiding mode collapse:** Our qualitative analysis on mode-collapse for toy datasets consistently shows how pruning helps us capture the individual modes in the data better. To support this study further, one can use the quantitative metrics proposed in [Srivastava et al. VEEGAN, NeurIPS 2017]: Method 1. Draw samples from the normalizing flow, and consider a sample of “good quality” if it is within 3 standard deviations from its nearest mode. We then can report the percentage of the good quality samples as a measure of how well the generative model captures modes. Method 2. Counting the number of captured modes: we can count the number of mixture components whose mean is nearest to at least one high-quality sample. Although we did not perform this experiment, in all 3 synthetic datasets (shown in Figure 5) we can observe qualitatively that both metrics are enhanced by applying a certain percentage of pruning. As the reviewer suggested, we will perform this quantitative analysis for all experiments, throughout the discussion period to support our claim.
>
> **Pruning flattens the loss surface:** Our Hessian-based robustness analysis is strong evidence for this statement. Moreover, we would like to refer the reviewer to a recent ICLR 2021 paper [Erichson et al. 2021 Lipschitz Recurrent Neural Networks], where a very similar Hessian-based robustness analysis is used to reason about the properties of the loss and gradient propagation in networks. Erichson et al. 2021 also picks one dataset to support their robustness of their method compared to others. This is a very common practice in quantitative justification of flat minima vs sharp minima (see for instance: [Hochreiter and Schmidhuber, Flat Minima, 1997, Li et al. ​​Visualizing the loss landscape of neural nets, 2017]. Moreover, virtually all of our observations scale consistently from smaller scale results to the larger scale benchmarks (including our observations for structural pruning vs unstructured pruning which we reported above). We believe it is not necessary to be shown on every single dataset presented in our paper. Though, if the reviewer still believes that it is essential to have the Hessian analysis performed for another dataset, we will provide results accordingly during the discussion period.
>
> **After a certain pruning ratio Hessian goes up. Provided evidence** We continued our Hessian analysis for a 90% pruned network, here are the Hessian values corresponding to that of Table 4:
>
> |Model|NLL|$\lambda_{max}(H)$|$\text{tr}(H)$|$\kappa(H)$|
> |---|---|---|---|---|
> |Unpruned FFJORD|1.309|1.000|1.000|1.000|
> |Sparse Flows (PR=20\%)|1.163|0.976|0.858|0.825|
> |Sparse Flows (PR=60\%)|1.125|0.356|0.583|0.717|
> |Sparse Flows (PR=70\%)|1.118|0.295|0.340|0.709|
> |Sparse Flows(PR=90\%)|1.148|0.416|0.366|0.547|
>
> We observe that with a 90% prune ratio, the Hessian started increasing which supports our claim.
>
> **A smaller FFJORD trained from scratch vs pruned FFJORD** As requested by the reviewer we trained a small FFJORD which has roughly the same size as a SparseFlow with 32K parameters (Prune Ratio = 96%) on the MINIBOONE dataset, and here are the results:
>
> | Model|nats|#param|
> |---|---|---|
> |Sparse Flow |9.95|32K|
> |FFJORD|10.50|821K|
> |Small FFJORD from scratch|23.1|33K|
>
> For the best comparison to obtain a small FFJORD, we took the big FFJORD and randomly removed 96% of its parameters to simulate a smaller FFJORD of the same capacity as a SparseFlow with 32K parameters.
>
> **Why experimenting with FFJORD and not CNFs?**
> FFJORD is an elegant trick to efficiently compute the log determinant of the Jacobians in the change of variables formula for CNFs. Thus, FFJORDs are CNFs but faster. Under these circumstances, would the reviewer still want to see additional experiments with CNFs on Toy data (note that CNFs are extremely slow as the size of the dataset increases, thus do not readily scale to our other experiments)?

---

> > ### Comment · Reviewer_LH4D · 2021-08-17
> > **Thanks for the detailed response!**
> >
> > Thank you for the detailed response, a large number of my points have been addressed. In particular, it is great to see the additional results already provided. I look forward to seeing more as they become available. Getting into the specifics...
> >
> > *Comparing structured and unstructured pruning:* Thanks for adding these tables and analysis. I think that these experiments are a great addition.
> >
> > *Key insights for designing better Neural ODE instances:* Thanks for the clarification regarding lines 299-306. I think an appendix section regarding these results and experiments would be very useful. In fact, I think that the statement "By providing key insights about how pruning improves generative modelling and inference, we enabled the design of better neural ODE instances" can only really be made in the main text if at least some of these experiments can also be found in the main text. Saying that "the first and foremost insight is the application of pruning itself to an ODE-based flow" is a bit disingenuous since this point has already been made in the same paragraph: "We showed the effectiveness of pruning for continuous neural networks. Pruning improved generalization performance of continuous normalizing flows in density estimation tasks at scale".
> >
> > *Avoiding mode collapse:* I think that this kind of quantitative experimentation would greatly strengthen the claim. The problem with qualitative evaluation on toy datasets is that they do not necessarily reflect the reality of complicated real-world datasets. I look forward to seeing these results.
> >
> > *Pruning flattens the loss surface:* I am not arguing with the Hessian based analysis nor do I think that this analysis must be done for all of the datasets in the paper. However, I think it is valuable to do the analysis on a few different datasets to confirm the generality of this claim. Please do provide some results when they are available.
> >
> > *After a certain pruning ratio Hessian goes up. Provided evidence:*  Thanks for the additional results! I am now happy with this. However, I think this point would be made stronger with a few more prune percentages. (And as a suggestion a plot instead of a table :))
> >
> > *A smaller FFJORD trained from scratch vs pruned FFJORD:* Thanks for adding this ablation. Would it be possible to have a few more datasets in the appendix of the camera-ready? (I am happy with this so no need to do this now.)
> >
> > *Why experimenting with FFJORD and not CNFs?:* I do appreciate that the choice of FFJORD was made for computational reasons. My question is whether or not the results generalise to normal CNFs. As a result, I would very much appreciate results for CNFs on toy datasets.

---

> > > ### Author Response · Authors · 2021-08-22
> > > **Thank you for your thorough and constructive review! Our work has improved significantly!**
> > >
> > >
> > > We would like to thank the reviewer very much for their thorough, fair, and truly constructive feedback on our manuscript. Indeed, this review has transformed our paper into a much stronger paper than the first iteration.
> > >
> > > We are also very happy to hear that most of the reviewer’s concerns have been addressed. In the following, we try to address the rest of the issues raised, and hope to have convinced the reviewer to vote for the publication of our results:
> > >
> > > **key insights for designing better Neural ODE instances:** We fully agree with the reviewer. We will move the design insights and our experiments in full in a dedicated Appendix section, and edit lines 299-306 to point at these results in the appendix.
> > >
> > > ### Avoiding mode Collapse
> > >
> > > We would like to thank the reviewer for motivating us to perform these experiments. As promised, we developed a quantitative measure adopted from [Srivastava et al. VEEGAN, NeurIPS 2017] for supporting our claims on the effectiveness of pruning on avoiding mode-collapse:
> > >
> > > Method: *The percentage of Good Quality samples---* We draw samples from a trained normalizing flow and consider a sample of “good quality” if it is within n (n is typically chosen to be 2, 3, or 5) standard deviations from its nearest mode. We then can report the percentage of the good quality samples as a measure of how well the generative model captures modes.
> > > The tables below summarize the results on the synthetic datasets and the large-scale MNIST dataset. We can see that the number of “good quality samples”, which stand for a quantitative metric for mode-collapse, improves with a certain percentage of pruning in all experiments.
> > >
> > > Moreover, this observation is very much in line with our results on the generalization loss. We will add this strong quantitative experiment to our revised manuscript to support our claims on pruning enhances mode-capturing in normalizing flows.
> > >
> > > **Synthetic Datasets for Mode-collapse Experiments (structured and unstructured pruning)**
> > >
> > > |Gaussian Mode-collapse Experiments |||
> > > |---|---|---|
> > > |**Structured Pruning** |||
> > > |  | **% of Good Quality Samples** ||
> > > | **Prune ratio (%)** |Std = 2 | std = 3 |
> > > | 0.0 | 72.89% |  89.34% |
> > > | 20.0 | 79.46% |  93.45% |
> > > | 25.6 |  81.40% |  94.90% |
> > > | 52.0 |  84.30% |  96.85% |
> > > | 71.2 |  81.83% |  94.86% |
> > > | 75.1 |  68.20% |  86.10% |
> > > | 98.0 |  36.01% |  76.54% |
> > >
> > >
> > > |Gaussian Mode-collapse Experiments |||
> > > |---|---|---|
> > > |**Unstructured Pruning** |||
> > > |  | **% of Good Quality Samples** ||
> > > | **Prune ratio (%)** |Std = 2 | std = 3 |
> > > | 0.0 | 72.78% |  89.41% |
> > > | 20.0 | 79.85% |  93.70% |
> > > | 25.6 |  82.73% |  95.60% |
> > > | 52.0 |  85.03% |  97.34% |
> > > | 71.2 |  85.31% |  97.57% |
> > > | 75.1 |  85.41% |  95.63% |
> > > | 98.0 |  30.01% |  72.65% |
> > >
> > > |Gaussian-Spiral Mode-collapse Experiments |||
> > > |---|---|---|
> > > |**Structured Pruning** |||
> > > |  | **% of Good Quality Samples** ||
> > > | **Prune ratio (%)** |Std = 2 | std = 3 |
> > > | 0.0 | 66.65% |  84.01% |
> > > | 20.0 | 74.29% |  90.64% |
> > > | 25.6 |  78.22% |  93.02% |
> > > | 52.0 |  80.40% |  94.29% |
> > > | 71.2 |  78.06% |  93.41% |
> > > | 75.1 |  78.99% |  93.67% |
> > > | 96.0 |  58.30% |  75.05% |
> > > | 98.0 |  9.27% |  20.32% |
> > >
> > >
> > > |Gaussian-Spiral Mode-collapse Experiments |||
> > > |---|---|---|
> > > |**Unstructured Pruning** |||
> > > |  | **% of Good Quality Samples** ||
> > > | **Prune ratio (%)** |Std = 2 | std = 3 |
> > > | 0.0 | 66.81% |  84.09% |
> > > | 20.0 | 77.76% |  92.63% |
> > > | 25.6 |  78.87% |  93.46% |
> > > | 52.0 |  81.77% |  95.26% |
> > > | 71.2 |  81.37% |  95.14% |
> > > | 75.1 |  81.80% |  95.30% |
> > > | 96.0 |  71.54% |  87.94% |
> > > | 98.0 |  69.56% |  78.82% |
> > >
> > > **MNIST Mode-collapse Experiments (structured and unstructured pruning)**
> > >
> > > |MNIST Mode-collapse Experiments |||
> > > |---|---|---|
> > > |**Structured Pruning** |||
> > > |  | **% of Good Quality Samples** ||
> > > | **Prune ratio (%)** |Std = 3 | std = 5 |
> > > | 0.0   |   6.94% |  58.24% |
> > > | 20.0 |   8.02% |  65.98% |
> > > | 37.8 |   **9.35%** |  **68.11%** |
> > > | 51.7 |   **8.70%** |  **67.24%** |
> > > | 62.4 |   **8.38%** |  **67.03%** |
> > > | 70.8 |   **8.50%** |  **66.56%** |
> > > | 77.3 |   **8.59%** |  **66.24%** |
> > > | 82.4 |   8.17% |  64.74% |
> > > | 86.3 |   6.95% |  60.45% |
> > > | 89.3 |   6.07% |  57.28% |
> > > | 91.7 |   0.02% |  32.75% |
> > >
> > >
> > > |MNIST Mode-collapse Experiments |||
> > > |---|---|---|
> > > |**UnStructured Pruning** |||
> > > |  | **% of Good Quality Samples** ||
> > > | **Prune ratio (%)** |Std = 3 | std = 5 |
> > > | 0.0   |   6.82% |  58.23% |
> > > | 20.0 |   8.30% |  64.79% |
> > > | 37.8 |   **8.95%** |  **67.67%** |
> > > | 51.7 |   **9.41%** |  **68.01%** |
> > > | 62.4 |   **8.63%** |  **68.92%** |
> > > | 70.8 |   **9.14%** |  **68.41%** |
> > > | 77.3 |   **8.27%** |  **67.29%** |
> > > | 82.4 |   **8.74%** |  **67.53%** |
> > > | 86.3 |   **8.63%** |  **67.13%** |
> > > | 89.3 |   8.07% |  64.33% |
> > > | 91.7 |   6.27% |  59.75% |
> > >
> > >
> > > ### Provide results for Hessian analysis on more datasets:
> > >
> > > We performed the following additional Hessian experiments:
> > >
> > > **Gaussians - Hessian Analysis (Structured Pruning)**
> > >
> > > |Model|NLL|$\lambda_{max}(H)$|$\text{tr}(H)$|$\kappa(H)$|
> > > |---|---|---|---|---|
> > > |Unpruned FFJORD      |1.173|0.0190|0.098|48.2k|
> > > |Sparse Flows(PR=25\%)|1.157|0.0110|0.076|2.76k|
> > > |Sparse Flows(PR=67\%)|1.148|0.0090|0.560|15.17k|
> > > |Sparse Flows(PR=82\%)|**1.120**|**0.0065**|0.058|22.75k|
> > > |Sparse Flows(PR=90\%)|**1.136**|**0.0035**|0.033|4.70k|
> > > |Sparse Flows(PR=94\%)|1.173|0.0069|0.033|3.94k|
> > > |Sparse Flows(PR=96\%)|1.244|0.0071|0.043|0.58k|
> > >
> > > **Gaussians-Spiral - Hessian Analysis (Structured Pruning)**
> > >
> > > |Model|NLL|$\lambda_{max}(H)$|$\text{tr}(H)$|$\kappa(H)$|
> > > |---|---|---|---|---|
> > > |Unpruned FFJORD      |0.880|0.0130|0.121|0.34k|
> > > |Sparse Flows(PR=25\%)|0.692|0.0076|0.058|0.76k|
> > > |Sparse Flows(PR=48\%)|**0.634**|**0.0049**|0.047|0.22k|
> > > |Sparse Flows(PR=67\%)|**0.646**|**0.0052**|0.051|0.75k|
> > > |Sparse Flows(PR=82\%)|**0.657**|**0.0053**|0.053|1.69k|
> > > |Sparse Flows(PR=94\%)|0.740|0.0086|0.070|0.11k|
> > > |Sparse Flows(PR=96\%)|0.986|0.0100|0.095|0.23k|
> > >
> > > We observe that our observations on the behavior of the Hessian is generalizable to other datasets as well. The results for the Spiral dataset is taking much longer and we will certainly include them in the camera-ready version.
> > >
> > > **Make a figure for Hessian analysis instead of the table** This is a great idea. This way, we can include more Hessian analysis in the paper without losing clarity of the presentation. We will do this in the camera-ready version.
> > >
> > >
> > > **I would very much appreciate results for CNFs on Toy datasets** As requested, we conducted these experiments with CNFs. The results are reported in tables below for each toy dataset (Except Spiral dataset which as we mentioned before faces significant computational overhead due to CNFs high time complexity, and is still running). We can see a very similar outcome to that of pruning FFJORD experiments, as expected:
> > >
> > >
> > > |**Gaussians - CNF**|||
> > > |---|---|---|
> > > |Network| Loss (NLL) | Parameter Count |
> > > |Unpruned CNF | 1.25|642|
> > > |Sparse Flows (WT, PR=20\%)| 1.17|512|
> > > |Sparse Flows (WT, PR=25\%)| 1.16|479|
> > > |Sparse Flows (WT, PR=52\%)| 1.15|307|
> > > |Sparse Flows (WT, PR=64\%)| 1.15|230|
> > > |Sparse Flows (WT, PR=75\%)| 1.16|160|
> > > |Sparse Flows (WT, PR=86\%)| 1.16|91|
> > > |Sparse Flows (WT, PR=90\%)| 1.21|65|
> > > |Sparse Flows (WT, PR=92\%)| 1.53|50|
> > > |Sparse Flows (WT, PR=94\%)| 2.07|39|
> > > |Sparse Flows (WT, PR=96\%)| 2.70|26|
> > > |Sparse Flows (WT, PR=98\%)| 2.82|13|
> > >
> > > |**Gaussians Spiral - CNF**|||
> > > |---|---|---|
> > > |Network| Loss (NLL) | Parameter Count |
> > > |Unpruned CNF | 0.78 | 8.64K  |
> > > |Sparse Flows (WT, PR=20\%) | 0.67 | 6.89K  |
> > > |Sparse Flows (WT, PR=26\%) | 0.62 | 6.43K  |
> > > |Sparse Flows (WT, PR=40\%) | 0.62 | 5.15K  |
> > > |Sparse Flows (WT, PR=55\%) | 0.63 | 3.87K  |
> > > |Sparse Flows (WT, PR=64\%) | 0.59 | 3.14K  |
> > > |Sparse Flows (WT, PR=75\%) | 0.60 | 2.19K  |
> > > |Sparse Flows (WT, PR=86\%) | 0.58 | 1.19K  |
> > > |Sparse Flows (WT, PR=92\%) | 0.60 | 691  |
> > > |Sparse Flows (WT, PR=94\%) | 0.61 | 522  |
> > > |Sparse Flows (WT, PR=96\%) | 0.79 | 347  |
> > > |Sparse Flows (WT, PR=98\%) | 1.44 | 164  |
> > >
> > > **A smaller FFJORD trained from scratch vs pruned FFJORD: Thanks for adding this ablation. Would it be possible to have a few more datasets in the appendix of the camera-ready? (I am happy with this so no need to do this now.)** Certainly. We will include more experiments in the appendix of the camera-ready version.
> > >
> > > With these additional instructions and extensions guided by the reviewer, we believe to have strongly improved our paper's claims and significance. We hope that the reviewer, as before, agrees with us and vote for the acceptance of our paper.

---

> > > > ### Comment · Reviewer_LH4D · 2021-08-22
> > > > **I agree**
> > > >
> > > > Thanks for providing the additional results and clarifications. I agree that the paper is now much stronger and so I am happy to increase my score.
> > > >
> > > > One final question, in figure 1 there is a purple line to show the trend in NLL when printing a FFJORD model. However, there is a second pink line that connects Glow, ResFlow and NSF, what is this one highlighting?

---

> > > > > ### Author Response · Authors · 2021-08-23
> > > > > **We are so grateful!**
> > > > >
> > > > > We would like to thank you very much for engaging so profoundly in the review process and guiding us towards making a strong contribution. Thank you!
> > > > >
> > > > > **Figure 1, pink highlight** With the pink highlight we wanted to provide a comparable projection range for other models to that of the trend obtained from pruning. The pink line is not to propose a connection between those other normalizing flows, but it is to visually guide how significantly smaller flows (sparse flows) can perform as well or better than their larger counterparts. If the reviewer thinks that the pink projection is misleading of any sort, we can omit it.
> > > > >
> > > > > ### Final Remarks
> > > > >
> > > > > As planned, executed, and promised, we commit to incorporate all changes in our camera-ready version, as we strongly believe this new set of experiments and insights drastically improves all aspects of our work such as significance, clarity, presentation, and impact.
> > > > >
> > > > > Thank you,
> > > > > Authors.

---

> ### Author Response · Authors · 2021-08-11
> **Response to Reviewer LH4D - PART 2 of 2**
>
> PART 2 of our response:
>
> ### Clarity Issues addressed:
>
> **1. 98% results explained**
> Please look at the smallest SparseFlow results on the MINIBOONE dataset in Table 2. We will clarify this further in the potential camera-ready version of our manuscript.
>
> **2. “Deciphering the internal dynamics of neural ODEs:”**
> The internal dynamics of a neural ODE is determined by the parametrized neural network f. By pruning those parameters we find out that the Hessian decreases ---> flatter local minima are obtained ---> better generalization in a diverse set of experiments. This is what we mean by deciphering the internal dynamics of Neural ODEs.
>
> **3. Clarification regarding the pruning procedure:**
> Please find below the requested clarifications. These will be incorporated into our revised manuscript:
>
> **3a. "inspired" by iterative learning rate rewinding**:
> Learning rate rewinding (LRR) is a hyperparameter schedule for training/pruning/retraining of discrete neural networks (see Renda et al., 2020). Essentially, we just wanted to clarify that our approach to pruning is related to Renda et al., 2020, among others. All the relevant aspects are mentioned in Algorithm 1).
>
> **3b. "pre-defined threshold" isn't specific enough**:
> We pick a desired prune ratio and prune the weights with the smallest magnitudes until we obtain the desired prune ratio. The largest weight that is being pruned constitutes the “pre-defined threshold” for pruning.
>
> **3c. it isn't clear exactly what is meant by "prune the structures" (what kinds of structures are being considered?), and**
> We consider neurons in fully-connected layers and filters in convolutional layers for structured pruning. The corresponding pruning score is the $\ell_1$-norm of the neuron/filter weights as specified in Table 1.
>
> **3d. it isn't clear what "we generally follow the same hyperparameters" means?**
> We do not change the training hyperparameters (optimizer, batch size, weight decay, etc..) between the different stages of training (Line and Line 6 of Algorithm 1)
>
> **4. Clarifying Figure 6:**
> Please read lines 197-205 in the manuscript.
>
> **5. What are the prune ratios in Table 2:** Table 2 represents a subset of Figure 8 which contains the prune ratios. The data points were chosen individually for each dataset to show a maximally diverse subset of prune ratios from Figure 8. Since the initial author notifications, we added additional prune ratios to this experiment to fill out the table.
>
> Complete Sparse Flow results of Table 2:
>
> | Power, nats & # params | Gas, nats & # params | Hepmass, nats & # params | Miniboone, nats & # params | Bsds300, nats & # params |
> | --- | --- | --- | --- | --- |
> | -0.45 & 30K | -10.79 & 194K | 16.53 & 340K | 10.84 & 397K | -145.62 & 4.69M |
> | -0.50 & 23K | -11.19 & 147K | 15.82 & 160K | 10.81 & 186K | -148.72 & 3.55M |
> | -0.53 & 13K | -11.59 & 85K | 15.60 & 75K | 9.95 & 32K | -150.45 & 2.03M |
> | -0.52 & 10K | -11.47 & 64K | 15.99 & 46K | 10.54 & 18K | -151.34 & 1.16M |
>
> **6. Why do figures 8 a), b), and e) not have the full range of pruning ratios?**
> See answer to 5.
>
> **7. Flows-under-test** means the flows we are testing as a baseline for pruning. In particular, it means CNFs supplied with FFJORD, unpruned and pruned.
>
> **8. Regarding Figure 9:**
>
> **8a. what does it mean "edgier"?**
> By an edgier vector field for the pruned network we mean the vector field bends less smooth than that of an unpruned version. As suggested, we will add some pointers in Figure 9.
>
> **8b. How does the attenuation of the vector field intensity indicate that the classification boundary is more sensitive to perturbations?**
> Let’s look at the first column and row 3 entry of Figure 9. We see that the classification boundary between the blue half-moon and the orange half-moon is very close to one tail of the blue half-moon. This means that if we perturb the blue half-moon data, some of them might get classified incorrectly based on the extremely pruned network’s decision boundary. As the intensity of the vector field shrinks to the center of the vector field (it is attenuated in the areas distant from the center), the decision boundary is set poorly as opposed to networks with more parameters.
>
> **8c. What should I be looking at in column 3 of figure 9? **
> Column 3 of Figure 9 shows the evolution of single samples (each narrow line represents an individual sample’s dimension 1 path) from input to their output through the depth of the network.
>
> **Additional Comments**
>
> 1. We agree and we will add the paper.
>
> 2. NSF is already benchmarked for CIFAR10 in Table 3 as RQ-NSF. However, for the other datasets (Tabular and MNIST), the paper does not provide the total number of parameters of the model. As our main focus was to compare the performance-size trade-off, we chose not to include NSF in the other benchmarks. Also, the NSF performance on Tabular and MNIST datasets is provided directly in the NSF paper which can be compared one to one to our experiments.
>
> 3. Thanks for the suggestion, we will update this.
>
> 4. We will add the following reference for BPTT: David E Rumelhart, Geoffrey E Hinton, and Ronald J Williams. Learning representations by back-propagating errors. nature, 323(6088):533–536, 1986.
>
> 5. Thanks, we will add the ground truth distributions to Figure 5.
>
> With this rebuttal, we hope to have addressed the major reviewer’s concerns. We look forward to hearing from the reviewer and are more than happy to address any additional clarifications requested.

---

> > ### Comment · Reviewer_LH4D · 2021-08-17
> > **Thanks for the detailed response - part 2!**
> >
> > I just have a few more follow-ups here.
> >
> > *4. Clarifying Figure 6:* I had read these lines, however, I still did not understand the figure. The problem is that I struggled to connect the text with the pictures. Hence my request for a more detailed explanation, some annotations, or an additional dataset. I understand that the annotations and additional dataset can't be supplied here, but a more detailed description, like the one provided for figure 8, would be great. It would also be great to know that there would be some additional care taken in the camera-ready.
> >
> > *6. Why do figures 8 a), b), and e) not have the full range of pruning ratios?* I'm afraid I still have this question. The answer for point 5 doesn't explain why these figures don't have the full range, especially since "Table 2 represents a subset of Figure 8". The reason I want the full range for a), b) and e) is that c) and d) show the turning point of the curves and it would be interesting to see the same for the other 3.
> >
> > *8c. What should I be looking at in column 3 of figure 9?* Sorry, I should have been more clear about this question. I know that the column shows the evolution of samples. What I want to know is what interesting trends I should be paying attention to. What is the purpose of this column?

---

> > > ### Author Response · Authors · 2021-08-22
> > > **Further Clarifications**
> > >
> > >
> > >
> > >
> > > **4. More detailed explanations on Figure 6**
> > >
> > > To further clarify Figure 6, we have prepared an additional annotated figure which can be found in the following anonymous link: https://figshare.com/s/720a735c4f07e8f86812
> > >
> > > In this figure, we show the distribution and vector field of learned FFJORD networks (unpruned (top) and 70% pruned down). The area in the vector field declared with a black circle shows the vector field structure around an actual mode in the dataset. We see that the vector field (which is illustrated by black arrows) attracts samples towards the mean of this distribution in both pruned and unpruned networks.
> > >
> > > However, there is a drastic difference between the vector field structure in-between modes (annotated by purple circles), between the unpruned and pruned network. In the unpruned network, the vector field attracts samples in-between modes. In contrast, in the pruned network, the vector field is repellent in-between modes. Correspondingly, this illustration shows how an unpruned network tends to have samples in-between modes, while the pruned network avoids this shortcoming.
> > >
> > > We hope this explanation clarifies this image further. We will add these annotations to our camera-ready version.
> > >
> > >
> > > **6. Show full curves for Figure 8 a b and e.**
> > >
> > > This is a great point to observe whether there is a turning curve for all experiments. Originally, we stopped the pruning experiments as we did not know where the turning point would occur and our objective was to show that pruning improves performance. We have now extended these experiments and provided their corresponding figures and data in the following anonymous link:
> > > https://figshare.com/s/b0fb3113371dad977128
> > >
> > > Now in all experiments (Figure 8), the turning point in the curve is illustrated. We will improve these further as the results become available and present them in our camera-ready version. Moreover, the link contains more results on structured vs unstructured pruning.
> > >
> > > **8c: What I want to know is what interesting trends I should be paying attention to. What is the purpose of this column?**
> > >
> > > The main observation we can get from Column 3 of Figure 9 is that when the networks are not pruned (column 3, row 1) or are pruned up to a certain threshold (column 3, row 2), the separation between the flow of samples of each class (orange or blue) is more consistent compared to the very pruned network (column 3, row 3). In Column 3, row 3 we see that the flow of individual samples from each class is more dispersed. As a result, the model is more sensitive to perturbations on the input space. This is very much aligned with our observation we have previously discussed the relation of column 1 and column 2 (explained in our previous response point 8b). We hope this explanation clarifies the purpose of Column 3 of Figure 9.

---

### Official Review · Reviewer_uTzg · 2021-07-16

**Rating:** 7
**Confidence:** 4

**Summary:**

The paper presents a finding that pruning improves generalisation for neural ODEs in generative modelling alongside with reducing the number of parameters to up to 98%.



**Ethical Concerns:**

No concerns

**Limitations And Societal Impact:**

The limitations of computational complexity and accuracy are thoroughly discussed. The societal impact relates to the general concerns about inappropriate use of machine learning methods and neural ODE based density estimation in particular.

**Main Review:**

*Pros:*

The proposed method is an attractive way of improving the number of parameters and at the same time regularising continuous generative models. The method has a potential for good impact for the neural ODE community and practitioners using the methods.

*Cons:*

Some of the description could be clarified (see below)

*Clarity:*

Line 132- 143: It would be good to tell  here how exactly this structured pruning method, local and global, is implemented for the continuous models, as the original methods are designed for (discrete) deep neural networks. Also, it would be good to define $W_ij$ in Table 1

*Ablation studies*

It looks convincing from the experiment that pruning has a good effect on performance, however, some additional ablation studies could help understand it further why it happens (decorrelation of parameters similar to the dropout, smaller number of parameters decreasing the chance of falling into undesirable local minima):
- how would the results compare with reducing the width of the dense network? If that helps too, it would suggest that just smaller number of parameters could improve learning.  I appreciate that it may or may not be feasible to disentangle it in the experimental setting from Neural ODE dimensionality as every continuous layer of neural ODE has the same dimension, however it would be good to hear the author's opinion on it in the response and in the paper.
- I wonder if keeping the number of parameters the same but introducing the analogue of dropout could help understand whether it gives the regularisation result similar to the dropout (and whether in that case the same number of parameters as in the original Neural ODE architecture would be beneficial).

*Other comments:*

- Table 2: Would it be possible to output confidence intervals as well?

- lines 299-306: the authors discuss the activation function choice, it would be good if the authors could clarify whether they have more evidence on its impact (and maybe put that evidence in the appendix).

== After rebuttal ==

As the authors sufficiently addressed the concerns in the rebuttal and carried out new ablation studies which could significantly improve the content of the paper, the score is increased from 6 to 7. However, as the reviewer could not open the main file with figures of the ablation study due to technical error and only could see the contents and conclusion of the ablation study, the confidence is decreased from 4 to 3.

**Update:** the confidence is increased back to 4 as the authors provided necessary support for ablation studies.


**Time Spent Reviewing:**

5

---

> ### Author Response · Authors · 2021-08-11
> **Response to Reviewer uTzg**
>
> We would like to thank the reviewer for their positive and constructive feedback on our manuscript. Please find our response in the following:
>
> ### Clarity issues
>
> **Lines [132-143] How pruning is applied to the continuous models:** ODE-based flows are determined by their right-hand side being a neural network $f$. We sparsify the flow, by applying standard pruning methods to the neural network $f$ (which is a discrete neural network). “Local” and “Global” indicate whether we prune per discrete layer (local) or globally across all discrete layers (global).
>
> **Define $W_{ij}$:** It simply denotes element $(i, j)$ of a matrix $W$, where we follow the convention that $y = Wx$, where $y$ and $x$ denote output and input of a fully-connected layer, respectively. $||W_{i:}||_1$ denotes the $\ell_1$-norm of row $i$ of $W$, i.e., the weights associated with output neuron $i$ of the corresponding fully-connected layer. We will add this to our revised manuscript.
>
> ### Ablation studies
>
> **How does reducing width of a network affect generalization?** ​​Thanks for denoting this study. Structured pruning precisely performs this suggested ablation study. We can see in experiments shown in Figure 4, how structured pruning (width reduction) leads to better generalization compared to an unpruned network, but is more sensitive to parameter reduction than an unstructured pruning framework. However, directly shrinking the network and trying to train the network from scratch would result in a lower accuracy compared to applying structured pruning to a large architecture. We confirmed this in our initial study on toy datasets (see below) but this is something that is frequently observed in the broader pruning literature.
>
> **The effect of Dropout:** Dropout has been used in almost all baselines we benchmarked SparseFlows against. A certain dropout rate (of around 0.1-0.2) improves the generalization of the flow, but still, the effect of dropout regularization is different from that of pruning. It is a great idea to perform such an experiment extensively where different regularization schemes are compared against each other. This will certainly be part of our continued effort.
>
> **Table 2:** Results presented in Table 2 coincide with the results shown in Figure 8. Their corresponding confidence intervals are illustrated in Figure 8 with shaded regions. We will include the numerical confidence intervals in the appendix in our revised manuscript. At the reviewer’s request, we are happy to provide these numbers during the discussion period.
>
> **lines 299-306:** We performed a lightweight ablation study early on to investigate whether pruning can provide valuable insights into architectural design choices for Neural ODEs. Specifically, we tested what is the maximal prune ratio for a given architecture without accuracy loss to understand what choice of architecture and hyperparameters provides a more robust design choice for Neural ODEs. The conclusion of that study is presented in Lines 299-306. We omitted these results from our manuscript to focus on the more relevant findings. As the reviewer requested, we will include these results and their detailed explanation in the appendix of the revised manuscript. If the reviewer would like to see the results on this during the discussion period, we are more than happy to provide an anonymous link to a preliminary version of these results.

---

> > ### Comment · Reviewer_uTzg · 2021-08-25
> > **Re: Response**
> >
> > Dear authors,
> >
> > Many thanks for the insightful responses which address the comments.
> >
> > Lines [132-143]: sounds good, would be good to change the description in the revision accordingly.
> >
> > *How does reducing width of a network affect generalization? ​​* I think this answer, together with the discussion with other reviewers, answers my questions.
> >
> > On Table 2, it makes sense now, I think it would be good to add them into the appendix.
> >
> > "If the reviewer would like to see the results on this during the discussion period, we are more than happy to provide an anonymous link to a preliminary version of these results." That would be quite useful to see, thank you for preparing it. If all of that looks good, I'll be happy to increase my score to 7.

---

> > > ### Author Response · Authors · 2021-08-28
> > > **Response**
> > >
> > > We would like to thank the reviewer for engaging with us during this discussion period and for their insightful comments and suggestions for improving our manuscript.
> > >
> > > **Ablation studies**
> > >
> > > We have prepared our systematic ablation study as requested by the reviewer and provided them in the following anonymized link:
> > > https://figshare.com/s/9988dbc2933a3273f19f
> > >
> > > Please find the detailed description of this ablation study in the Readme.txt file. We include this file's description here as well:
> > >
> > > #  DESIGN NOTES FROM PRUNING NEURAL ODEs  #
> > > *An ablation study over different types of features of neural ODE models*
> > >
> > > We test different configurations by applying our SparseFlows (Algorithm 1) to investigate what type of network configurations are most stable and robust with respect to pruning. For each type of **sweep (ablation)**, we highlight one key study and one key result:
> > >
> > > ### SWEEP OVER OPTIMIZATION PARAMETERS
> > >
> > > **Setup:**
> > > We study the stability of different configurations for the optimizer and how the different configurations affect the generalization performance during pruning.
> > >
> > > **Key experiment:**
> > >   classifications_moon/opt_sweep_unstructured
> > >
> > > **Key observation:**
> > > We can find the most stable parameter configuration for the optimizer by considering sparsifying the flow and thus inducing additional regularization. The most stable optimizer configuration is the one for which we can achieve the most pruning.
> > >
> > >
> > > ### SWEEP OVER MODEL SIZES - DEPTH VS. WIDTH
> > >
> > > **Setup:**
> > >   We study different network configurations with (approximately) the same number of parameters. The networks differ in the depth vs. width configuration. We test deep and narrow vs. shallow and wide.
> > >
> > > **Key experiments:**
> > >
> > > ffjord_gaussians/model_sweep_unstructured
> > >
> > > ffjord_spirals/model_sweep_unstructured
> > >
> > >
> > > **Key observation:**
> > >   Increasing the depth of the network while reducing the width of the network, in general, does not help improve the generalization performance of the network over different prune ratios. Specifically, one should pick the minimal depth of the RHS that ensures convergence. Usually, any depth beyond that does not help improve the generalization performance of the flow.
> > >
> > > ### SWEEP OVER ACTIVATIONS
> > >
> > > **Setup:**
> > >   We study the same network configurations for the same amount of pruning and vary the activation function of the neural network on the RHS. As we prune, we hope to unearth which activation function is most robust to pruning and consequently to changes in the architecture.
> > >
> > > **Key experiment:**
> > >   ffjord_gaussians/activation_sweep_unstructured
> > >
> > > **Key observations:**
> > >   ReLU is usually not a very useful activation function. Rather, some Lipschitz continuous activation functions are most useful. Generally, we
> > >   found *tanh* and sigmoid to be most useful, although sigmoid was probably the most robust single configuration across all experiments
> > >
> > >
> > > ### SWEEP OVER ODE SOLVERS
> > >
> > > **Setup:**
> > > We study the same network configurations for the same amount of pruning and vary the ODE solver of the neural ODE flow. As we prune, we hope to unearth which solver is most robust to pruning and consequently to changes in the architecture.
> > >
> > > **Key experiment:**
> > > ffjord_gaussians/solver_sweep_unstructured
> > >
> > > **Key observations:**
> > > Generally, we found adaptive step size solvers (dopri5) superior to fixed step size solvers (rk4, Euler). Moreover, we found backpropagation through time (BPTT) to be slightly more stable than the adjoint method. Interestingly enough, we could oftentimes only observe the differences between the robustness of the different solvers after we start pruning and sparsifying the flows.
> > >
> > >
> > >
> > > As promised and agreed, we will add these results as an appendix to our paper. We hope that the reviewer is now happy with these results.

---

> > > > ### Comment · Reviewer_uTzg · 2021-08-30
> > > > **Description is good, cannot open the file due to technical error**
> > > >
> > > > The contents of ablation studies look good (especially given that the authors address the request fro ablation studies with varied network width) .  Unfortunately, I am not able to open the file https://figshare.com/s/9988dbc2933a3273f19f due to some technical error ( after clicking 'Download' returns the following text: {"message": "Entity not found: file", "code": "EntityNotFound"} ).
> > > >
> > > > Given that we do not have much time at the moment and I cannot expect the authors to send me the code, I thought that I should increase the score to 7  (but decrease confidence to 3 as I do not know what the exact figures of this ablation study are, and how it fits into the resulting paper). However, if the authors could update the link, I'm happy to have a look at it and if it meets the expectations, change the confidence.

---

> > > > > ### Author Response · Authors · 2021-08-30
> > > > > **Apologies for the broken link!**
> > > > >
> > > > > Thank you very much for letting us know about the issue. Anonymous link sharing is getting really tricky.
> > > > > Please find the correct link here:
> > > > >
> > > > > https://figshare.com/s/17a1a3aedd4c7b9e034e
> > > > >
> > > > > Please let us know if you could successfully access the data.
> > > > >
> > > > > Thanks

---

> > > > > > ### Comment · Reviewer_uTzg · 2021-08-30
> > > > > > **Received**
> > > > > >
> > > > > > Thank you for your prompt response; received, having a look and will let you know soon.

---

> > > > > > > ### Author Response · Authors · 2021-08-30
> > > > > > > **Great!**
> > > > > > >
> > > > > > >
> > > > > > > Thank you for the confirmation. We look forward to receiving your inputs on these experiments.

---

> > > > > > > > ### Comment · Reviewer_uTzg · 2021-08-31
> > > > > > > > **I had a look**
> > > > > > > >
> > > > > > > > I had a look at the experiments results, I think it shows significant improvement of the analysis as detailed in the authors' description above and justifies my score. I increase the confidence back to 4 and still recommend acceptance (7).

---

> > > > > > > > > ### Author Response · Authors · 2021-08-31
> > > > > > > > > **Thank you so much!**
> > > > > > > > >
> > > > > > > > >
> > > > > > > > > We would like to thank you very much for engaging with us during this fruitful discussion and for your fair and constructive review of our manuscript.
> > > > > > > > >
> > > > > > > > > Thank you,
> > > > > > > > > Authors

---

### Official Review · Reviewer_nNvs · 2021-07-22

**Rating:** 6
**Confidence:** 5

**Summary:**

**Summary**

The paper describes the effect of using pruning during neural ODEs training. Namely, the authors propose to train neural ODEs iteratively by alternating training (fine-tuning) and pruning steps. They demonstrate the benefits of applying the proposed training procedure to the density estimation tasks. Also, the paper considers a simple classification task to compare the robustness of decision boundaries of pruned ODE-based flows.

**Limitations And Societal Impact:**

Concerning limitations, there would be nice to have more information about time complexity (see Questions).

**Main Review:**

**Originality**

Training procedure that alternates pruning and fine-tuning steps is not new. The novelty of the proposed paper is application of this idea to neural ODEs.  The paper investigates the properties of pruned (sparse) neural ODEs. They provide examples to show how the usage of pruning (structural/unstructural) improves generalization, flattens the loss surface, and helps to avoid mode-collapse. They make an overview of important papers related to generative modelling and pruning.

**Quality**

I'd consider this paper as a complete piece of work that requires some clarifications. There are several questions concerning empirical experiments that should be specified (please, see the Questions section below). Also, I'd recommend writing a passage about time computational costs (both wall clock and number of function evaluations) during training/inference required for sparse neural ODEs comparing to dense neural ODEs.

**Clarity**

Both background and method description are clearly written. I'd suggest adding the following paper when overviewing efficient neural ODEs training: (Daulbaev, T. et al. Interpolation Technique to Speed Up Gradients Propagation in Neural ODEs. 2020)

**Significance**

Though the training with alternating pruning/fine-tuning steps is not new, its application to neural ODEs leads to several benefits, particularly significant improvement in parameters/loss trade-off for density estimation tasks.

**Questions**
- What are time costs (wall clock and number of function evaluations) during training/inference?
- Suppose we start with two neural ODEs of different layer widths and end up with two models of the same number of parameters. What difference we'll see in the properties of the resulting models? (If there is any)
- What is a motivation to use different optimizers in experiments (sometimes Adam, sometimes AdamW)?
- What induces the bigger difference for structural and unstructural pruning in the case of Gaussian distribution? (Figure 4)



**Time Spent Reviewing:**

5

---

> ### Author Response · Authors · 2021-08-11
> **Response to Reviewer nNvs**
>
> We would like to thank the reviewer for their positive evaluation of our work and their constructive feedback on our manuscript. Please find our response in the following:
>
> ### Questions
>
> **Time Costs Analysis:**
> We initiated an experiment for computing the number of function evaluations for sparsified and unpruned networks. Our preliminary observations show that the NFEs go up as we prune more, at the same time, however, the computational complexity per function evaluation reduces (measured in terms of FLOPs). We note that overall, the decrease in FLOPs per function evaluation (F/FE) dominates the increase in NFEs. In other words, the total computational complexity measured in terms of total FLOPs, i.e., NFE $\times$ F/FE, decreases as we prune more, which is a desirable outcome. We will add this analysis as a formal discussion to our revised manuscript. In case the reviewer would like to see the final results of our experiment, we will provide them during the discussion phase.
>
> **Suppose we start with two neural ODEs of different layer widths and end up with two models of the same number of parameters. What difference will we see in the properties of the resulting models? (If there is any)**
> This is an interesting scenario. Based on our experience, and depending on the dataset at hand, we speculate that both architectures (as long as they do not differ by multiple orders of magnitude) will end up with the same best performance vs pruning ratio. This has to be more rigorously studied though.
>
> **Motivation of the use of different optimizers:**
> We chose the optimizer suggested by the baselines we compared against, as they have given rise to better performance for the particular task-under-test.
>
> **Difference between structured and unstructured pruning in case of Gaussian distribution (Figure 4):** In general, structured pruning is a more constrained problem as we constrain the type of sparsity. Hence, for almost any pruning experiment we can expect that structured pruning performs “worse” than unstructured pruning starting at a certain prune ratio.
> In the particular case of Figure 4, data representation is most likely the prominent reason for the apparent difference between the different datasets. In Figure 4a, unstructured and structured pruning require about 15% and 35% of parameters, respectively, for the smallest model without a significant increase in loss. In Figure 4b, unstructured and structured pruning requires about 10% and 20% of parameters, respectively, for the smallest model without a significant increase in loss. In both cases, this implies that structured sparsity roughly requires 2x-3x more parameters than unstructured sparsity. In that sense, our observations are consistent across datasets. Moreover, we use a different architecture for GAUSSIANS and GAUSSIANSPIRAL (see Table S1 in the appendix). Therefore, the observations do not necessarily carry over 1:1.
>
> **Clarity**
> Thanks for denoting this paper (Daulbaev, T. et al. Interpolation Technique to Speed Up Gradients Propagation in Neural ODEs. 2020). We will make sure to include it in a potential camera-ready version of our work.

---

### Decision · Program_Chairs · 2021-09-27

**Decision:**

Accept (Poster)

**Comment:**

The following is a summary of the pros and cons raised by the reviewers:

pros:
* complete work (nNvs)
* Method and background sections are clearly written. (nNvs, AFJg)
* application of pruning to neural ODEs leads to several practical benefits such as generalization and reduced parameter count and its simplicity is a plus (nNvs, uTzg, LH4D)

cons:
* Requires additional experimental results:
    - wall-clock time and number of function evalutions during training and inference should be reported to see how pruning influences
this. --> preliminary results described in rebuttal, no reply from reviewer. (nNvs)
    - More evidence is needed to support claims such as flattening the loss surface. --> addressed during discussion period. (LH4D)
    - missing ablation studies that could help improve understanding of why pruning has benefits --> addressed in rebuttal.(uTzg)
    - structured vs unstructured pruning should be investigated on larger scale experiments, not only toy datasets --> addressed during discussion period. (LH4D)
    - pruning should be applied to other continuous normalizing flows, not just ffjord. --> authors have provided results on other cnf's on a toy dataset. (AFJg, LH4D)
* clarifications are required with respect to the pruning strategy and optimization choices. --> addressed in rebuttal (nNvs, uTzg)
* findings are unsurprising (AFJg).

The authors and 3 out of 4 reviewers engaged in active and fruitful discussions during the discussion period. The authors included many additional results that were requested by the reviewers, which led to the main concerns being addressed and 3 out of 4 scores being raised. The recommended decision for this submission is accept.